# Effects of Imazapyr on *Spartina alterniflora* and Soil Bacterial Communities in a Mangrove Wetland

Xue Mo [1], Panpan Dong [1], Lumeng Xie [1], Yujiao Xiu [1], Yanqi Wang [1], Bo Wu [2], Jiakai Liu [1], Xiuhua Song [3], Mingxiang Zhang [1,*] and Zhenming Zhang [1,*]

1 College of Ecology and Nature Conservation, Beijing Forestry University, Beijing 100083, China; mxoue998@163.com (X.M.); dpp1174724319@163.com (P.D.); xielumeng@bjfu.edu.cn (L.X.); xiuyujiao0111@bjfu.edu.cn (Y.X.); wangyanqi199806@163.com (Y.W.); timberfield1991@163.com (J.L.)
2 Beijing Top Green Ecological Technology Limited Company, Beijing 100005, China; bob1880@163.com
3 Management and Service Center of Huangshui National Wetland Park, Xining 810008, China; mingzhzh010@126.com
* Correspondence: zhangmingxiang@bjfu.edu.cn (M.Z.); zhenmingzhang@bjfu.edu.cn (Z.Z.)

**Abstract:** The invasion of *Spartina alterniflora* (*S. alterniflora*) has caused serious damage to coastal wetland ecosystems in China, especially the mangrove wetlands in South China. This study aimed to validate the effect of imazapyr on *S. alterniflora* and soil. The controlled experiment was conducted in May 2021 at the Zhangjiangkou Mangrove Wetland Reserve. In the experiment, 25% (W) imazapyr was used, and six treatments were set up: 3035, 6070, and 9105 mL/acre 25% imazapyr and 1299, 2604, and 5202 mL/acre of AGE 809 + 6070 mL/acre 25% imazapyr. The results showed no side effects on mangrove plants in the spraying area. The highest control efficiency (95.9%) was given by 2604 mL/acre of AGE 809 + 6070 mL/acre 25% imazapyr. The residues of imazapyr in different soils were reduced to 0.10–0.59 mg/kg. The sequencing results showed no significant difference in the overall bacterial communities under different treatments (*p* > 0.05). The soil bacterial diversity in the samples with adjuvant was higher than that in the samples without adjuvant, while the abundance values were the opposite. There were 10 main communities (>0.3%) at phylum level in all soil samples, among which Proteobacteria, Bacteroidetes, Acidobacteria, Chloflexi, and Actinobacteria were the dominant communities, and the latter four's abundance changed significantly (*p* < 0.05). There were significant abundance differences between the groups of oligotrophic and eutrophic bacteria. The redundancy analysis and Monte Carlo tests showed that the total organic carbon (TOC), total phosphorus (TP), available phosphorus (AP), ammonia nitrogen, and total nitrogen were the main factors affecting soil bacterial diversity. At the same time, TOC, AP, and TP were the most critical factors affecting the overall characteristics of soil bacterial communities in different treatments, while soil residues had no significant effect on bacteria. This might be due to the addition and degradation of imazapyr and the coverage of *S. alterniflora*. The best recommendation is 2604 mL/acre of AGE 809 + 6070 mL/acre 25% imazapyr to be applied in China's mangrove wetland reserves and coastal wetlands.

**Keywords:** imazapyr; mangrove wetland; soil bacterial diversity and community structure; soil physicochemical properties; *Spartina alterniflora*

## 1. Introduction

Alien species seriously threaten local ecosystems throughout the world by decreasing biodiversity, destroying the integrity of habitats, and reducing ecological functions [1–4]. *Spartina alterniflora* has drawn widespread attention as a global invasive species [5]. In 1979, *S. alterniflora* was introduced from the Atlantic coast of North America to China to slow down coastline erosion, protect beaches and dikes, and promote land siltation [6,7]. However, because of its strong adaptability and superior competitiveness, its population expanded rapidly in China's coastal areas from Liaoning Province in the north to the Leizhou

Peninsula in the south [8]. This caused the biomass and biodiversity decline of native plant species [9–11], such as mangrove forests with abundant soil carbon storage and numerous ecosystem services, which are among the most productive forests in the tropics [12]. Under the stress of *S. alterniflora*, the ecosystem structure changed, which led to functional degradation. The mangrove forest area sharply declined, and the Chinese government highlighted its potential effects and required urgent effective restoration methods [1,8].

In China, physical control methods are primarily used to control *S. alterniflora*. These include various combinations of rhizome removal, deep turning and landfill, incineration, harvesting, waterlogging, and shading [13–15]. However, these methods are expensive and have poor operability and low efficiency [15,16]. Such methods cause potential side effects to ecological processes and threaten environmental security. In contrast, the main treatment methods abroad are chemical control methods. These include the removal of *S. alterniflora* using herbicides, such as glyphosate [17], haloxyfop-R-methyl [18,19], and imazapyr [20]. Among these, imazapyr is the herbicide widely used abroad with the least environmental risk [21].

Imazapyr is a broad-spectrum imidazolinone herbicide used to control many types of grass, broadleaf weeds, and woody species with high efficiency and low toxicity. Furthermore, tidal inundation over canopies has little impact on its control performance [21], making it suitable for eliminating *S. alterniflora* along coastlines. After 10 years of extensive use of imazapyr on its west coast, the United States has achieved very effective results. It has controlled the invasion of *S. alterniflora*, and subsequent environmental monitoring studies showed that the use of the herbicide had no risk to the surrounding environment and organisms [20,21].

Soil microorganisms are the most active part of the soil [22]. They directly or indirectly participate in soil formation, decomposition of organic matter, nutrient cycling, the absorption of soil resources by plant roots, and seed germination [23,24]. Aside from their important roles in terrestrial ecosystem functions, soil microorganisms are sensitive to soil environmental changes. Thus, they can be used as indispensable and essential biological indicators for soil quality evaluation. Their composition and quantity can reflect soil quality, and their diversity and community structure can effectively reflect the stability of the soil system [25]. Therefore, the impact of herbicides on soil microbial communities must be studied.

So far, there are no reports on the use of imazapyr to control *S. alterniflora* in China; its ability to prevent and control the invasion of *S. alterniflora* in China, and its possible environmental impacts are unclear. In addition, the USA has been studying the effect of imazapyr removal of *S. alterniflora* on water, non-target plants, birds, insects, and fish for many years, but not the effects on soil physicochemical properties and microorganisms [26]. Therefore, the aim of this study is: (1) to analyze the control effect of imazapyr and its mixture with adjuvant on *S. alterniflora* in Chinese wetlands; (2) to evaluate the effect of imazapyr on soil, especially on soil microbial communities and dominant factors; and (3) to screen the best formula of imazapyr: the most efficient, economical, and environmentally friendly. This study can provide a scientific basis for policy formulations for controlling *S. alterniflora* and practical guidance for effectively controlling the harm of *S. alterniflora* to the environment. It can promote the protection and restoration of coastal wetlands and provide a scientific basis for the wide application of imazapyr.

## 2. Materials and Methods

### 2.1. Overview of the Study Area

The study was conducted in Zhangjiangkou Mangrove Wetland Reserve, Yunxiao County, Fujian Province (Figure 1; 117°24′07″–117°30′00″ E, 23°53′45″–23°56′00″ W). Zhangjiang Estuary Mangrove Wetland Reserve experiences a subtropical maritime monsoon climate, which is warm and humid. The annual average temperature is 21.2 °C, the extreme maximum temperature is 38.1 °C, and the extreme minimum temperature is 0.2 °C, the annual average precipitation is 1714.5 mm, the annual evaporation is 1718.4 mm. It is

mainly a mangrove wetland ecosystem which is rich in species, including 154 bird species and more than 400 species of aquatic creatures. Zhangjiang Estuary Mangrove Wetland Reserve is located at the intersection of saltwater and freshwater at Zhangjiang Estuary, and the most important wetland type is permanent mangrove wetland, with mangrove plants such as *Kandelia candel*, *Aegiceras corniculatum*, and *Avicennia marina* as constructive species. The soil in this area primarily comprises coastal beach silt and sandy silt, and is considered salinized soil with fine matrix particles, with oxygen deficiency, a reductive state, strong acidity, and organic matter. The experimental area is located in the intertidal tidal flat, the plant community in the area is completely submerged during high tides and the low tide is 5–8 h (the main time for plants to absorb imazapyr). During low tide, the sea water cannot submerge the leaves of *S. alterniflora*.

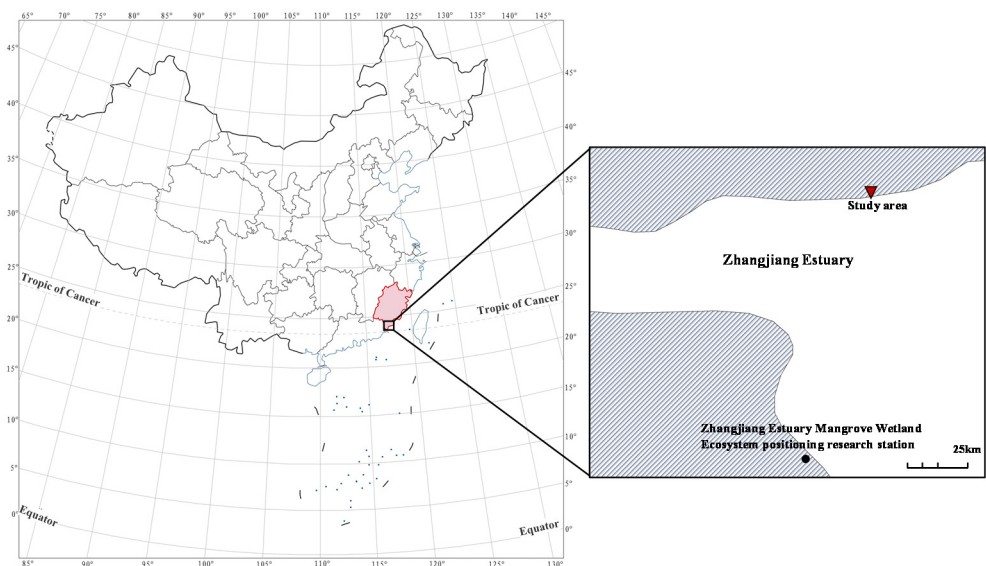

**Figure 1.** Study area (Zhangjiang Estuary Mangrove Wetland Reserve).

### 2.2. Site Setting and Experimental Methods

The sample sites were located at 117°25′07″ E, 23°56′30″ W, in May 2021 during the growing stage of *S. alterniflora*. A 1 ha study area was delineated where *S. alterniflora* was densely distributed. A total of six sites were set in this area, with three parallel sites (4 m × 10 m) for each site, totaling 18 sites with a combined area of 720 m². Sufficient spacing was set between the sites. Each site had one treatment (Table 1), and the sites were numbered Ia, Ib, Ic, IAa, IAb, and IAc.

**Table 1.** Agents and treatment methods.

| Tested Herbicide | Active Ingredient Concentration | Absorption/Action Site | Production Company |
|---|---|---|---|
| 25% imazapyr water agent | 22.62% | Leaves | Beijing Top Green Ecological Technology Co., Ltd. |
| AGE 809 (Lecithin) | 50% | Leaves | Beijing Top Green Ecological Technology Co., Ltd. |
| Sample number | Treatment method | | |
| Ia | 25% imazapyr 3035 mL/acre | | |
| Ib | 25% imazapyr 6070 mL/acre | | |
| Ic | 25% imazapyr 9105 mL/acre | | |
| IAa | 25% imazapyr 6070 mL/acre + AGE 809 1299 mL/acre | | |
| IAb | 25% imazapyr 6070 mL/acre + AGE 809 2604 mL/acre | | |
| IAc | 25% imazapyr 6070 mL/acre + AGE 809 5202 mL/acre | | |

Imazapyr spraying was conducted using an sprayer (SX-MD16E, SeeSa, Zhejiang, China) with a knapsack fan-shaped sprinkler head. After spraying, the number per square meter of plants (plant density) was regularly monitored. The sampling time covers the day before spraying and Days 15, 30, and 60 after spraying imazapyr, samples were collected after tide-ebbing. The aboveground biomass of the *S. alterniflora* community was investigated on Day 30 after spraying imazapyr. The collection frequency of soil and plant samples covers the day before spraying and Days 1, 6, 30, and 60 after spraying imazapyr.

In each parallel site, three sample areas covering 0.25 m$^2$ were randomly set, and the number of *S. alterniflora* was recorded. Soil samples were randomly collected from 0–20 cm of the soil of each parallel site. The soil collected from the three areas of each treatment was finally mixed into one sample and labeled with the site number (Ia$n$, Ib$n$, Ic$n$, IAa$n$, IAb$n$, and IAc$n$ at Days 1, 6, 30, and 60 after spraying imazapyr). After removing impurities such as roots and stones, some soil samples were taken out and stored in dry ice buckets. They were then transported back to the laboratory at low temperature and stored in an environment of $-80\ °C$ for microbial sequencing. The remaining part of the soil and other samples were divided into two parts. One part was stored at $-20\ °C$ for determining ammonia-nitrogen ($NH_3$-N), nitrate-nitrogen ($NO_3^-$-N), and imazapyr residues. The other part was ground and sieved through a 100-mesh after air drying to determine the pH, total carbon (TC), total nitrogen (TN), total phosphorus (TP), total organic carbon (TOC), and available phosphorus (AP). Several collected *S. alterniflora* leaves of each sample were stored at $-20\ °C$ and sent to the laboratory to determine imazapyr residue.

### 2.3. Determination Methods of Residues and Soil Physicochemical Properties

Imazapyr residues in soil and plants were determined according to GB/T 27860-2011 and the high-efficiency liquid chromatography method, respectively. The soil moisture content (VWC) and electrical conductivity (EC) were directly measured by a soil recorder (RS-TRREC-N01-1, Renke, Shandong, China) on site. TC was determined using a TOC analyzer (vario TOC SELECT, Elementar, Shanghai, Germany). TP was determined using the Kjeldahl distillation method. Referring to HJ 615-2011, TOC was determined by the potassium dichromate oxidation spectro-optical method. AP was determined by the sodium bicarbonate dipping–cyclic anti-spectrophotometric method referring to HJ 704-2014. In terms of NY_T 1121.2-2006, a pH/OPR/electrical conductivity/dissolved oxygen measuring instrument (SX751, SANXIN, Shanghai, China) was used to measure pH. Following LY/T1232-2015, TP was determined using the sodium hydroxide melting–ultraviolet-visible spectrophotosis method. Referring to HJ634-2012, the $NH_3$-N and $NO_3^-$-N contents were respectively determined using the KCl dip-double wavelength UV color comparison method and KCl dip-indigo-phenolic color comparison method.

### 2.4. Soil DNA Extraction, PCR Amplification, and Sequencing

The microbial community DNA was extracted using the NucleoSpin Soil Kit (Macherey-Nagel, Shenzhen, Germany) following the manufacturer's instructions. DNA was quantified with a Qubit Fluorometer using a Qubit dsDNA BR Assay kit (Invitrogen, Zhejiang, USA), and the quality was checked by running an aliquot on 1% agarose gel.

Thirty nanograms of qualified genomic DNA samples and corresponding fusion primers were taken to configure the polymerase chain reaction (PCR) system. Then, PCR parameters were set for PCR amplification. Variable regions V3–V4 of the bacterial 16S rRNA gene was amplified using degenerate PCR primers: 341F (5′-ACTCCTACGGGAGGCAGC AG-3′) and 806R (5′-GGACTACHVGGGTWTCTAAT-3′). PCR enrichment was performed in a 50-μL reaction containing the 30-ng template, fusion PCR primer, and PCR master mix. The PCR cycling conditions were as follows: 94 °C for 3 min, 30 cycles at 94 °C for 30 s, 56 °C for 45 s, 72 °C for 45 s, and final extension for 10 min at 72 °C. The PCR products were purified using Agencourt AmpureXP beads and dissolved in an elution buffer. Libraries were qualified using the Agilent 2100 bioanalyzer (Agilent, Shenzhen, USA). The validated

libraries were used for sequencing on the Illumina MiSeq platform (BGI, Shenzhen, China) following the standard pipelines of Illumina and generating $2 \times 300$ bp paired-end reads.

Off-machine data were filtered, and the remaining high-quality clear data were used for post-analysis. We stitched reads into tags through the overlapping relationship between reads. Tags were clustered into operational taxonomic units (OTUs) and compared with the library, and the species were annotated. The sample species complexity analysis, intergroup species difference analysis, correlation analysis, and model prediction were based on the OTUs and annotation results.

*2.5. Data Statistics and Analysis*

$$
\begin{aligned}
&\text{Plant density control effect rate} (\%) \\
&= \frac{\text{Number of plants in the control area} - \text{Number of plants in the application area}}{\text{Number of plants in the control area}}
\end{aligned}
\tag{1}
$$

SPSS 22.0 was used for the one-way analysis of variance (ANOVA) analysis and Pearson correlation analysis to analyze the significance of the differences and correlations of plant characteristics, imazapyr residue, soil physicochemical properties, and microbe-related data under different treatments. At the same time, QIIME was also used for cluster analysis (OTUs), alpha diversity index analysis (mainly including the Chao1, Shannon, Simpson, and ACE indices), and weighted-UniFrac principal coordinate analysis (PCoA). The R package version 3.1.1 was used for plotting the species accumulation curve, hierarchical cluster analysis, and heatmap production of the microbial genus composition. Furthermore, Canoco 4.5 was used for redundancy analysis (RDA) to further explore the main driving factors of the overall change in the main soil microbial communities in the different soil samples.

## 3. Results and Discussion

### 3.1. Effects of Imazapyr on Plants

As an alien species, *S. alterniflora* is known for its high yield, high resistance, and high reproduction. Figure 2 shows the control effects on the plant density under the six treatments. As can be seen from the figure, there was no significant difference between the treatments for each group for 15 days. After 15 days, the control effect of imazapyr alone was greater than those of imazapyr and AGE809, which gradually increased with time. The control rate after 30 days was significantly higher than that after 15 days, and the maximum value reached 60.38%. After 60 days, the control rates of each treatment had little difference, reaching more than 90%, and the highest was that of IAc (up to 95.9%). The result was higher than the control efficiency in most foreign studies on the removal of *S. alterniflora* by imazapyr, which is usually about 70% [21]. Moreover, it was shown that imazapyr removed *S. alterniflora* more efficiently when it is mixed with an adjuvant. The adjuvant AGE 809 has lecithin as its main component, which plays an important role in regulating physiological functions [27]. Water droplets containing a surfactant will spread in a thin layer over a waxy leaf surface and improve herbicide uptake by improving herbicide distribution on the leaf surface. This is of great significance for removing *S. alterniflora*.

### 3.2. Effects of Imazapyr and Its Additives on the Physicochemical Properties of the Topsoil

The physicochemical properties of the soil (Figure 3) showed that only the changes in TN content were significant in different treatment groups ($p < 0.05$). TP, TOC, AP, $NH_3$-N, and $NO_3^-$-N had significant differences in different periods. The TN content in the soil with 9105 mL/acre 25% imazapyr per acre was the highest. With a decrease in the 25% imazapyr addition level, the TN content decreased, indicating that the residual TN in soil may be related to the addition level of imazapyr. Meanwhile, the other soil properties did not change significantly. The highest values for the volumetric VWC, EC, pH, TC, TP, TOC, AP, $NH_3$-N, and $NO_3^-$-N were those of Ia, IAc, IAc, IAa, Ic, Ic, Ib, Ib, and Ia, respectively. The values of all soil physicochemical properties were lower for the low-concentration

adjuvant treatments, which can be attributed to the fact that the main component of the adjuvant is lecithin.

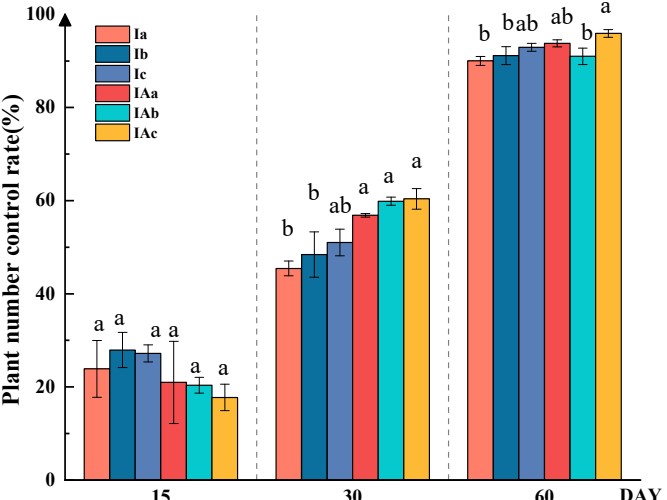

**Figure 2.** Plant density control rates under different treatments at different times. The different lowercase letters indicate significant differences between the treatment groups ($p < 0.05$). Ia: 25% imazapyr 3035 mL/acre; Ib: 25% imazapyr 6070 mL/acre; Ic: 25% imazapyr 9105 mL/acre; IAa: 25% imazapyr 6070 mL/acre + AGE 809 1299 mL/acre; IAb: 25% imazapyr 6070 mL/acre + AGE 809 2604 mL/acre; IAc: 25% imazapyr 6070 mL/acre + AGE 809 5202 mL/acre 25% imazapyr.

The imazapyr soil residues (Figure 4) in Ia and Ib were significantly lower than those in the other samples on Days 30 and 60. The soil residues in all quadrats were 0.13, 0.26, 0.83, 0.57, 0.67, and 0.59 mg/kg, respectively, after the 30th day and 0.10, 0.13, 0.26, 0.25, 0.26, and 0.59 mg/kg, respectively, after 60 days. The sample treated with high-dose imazapyr with high-dose adjuvant had the highest values ($p < 0.05$), indicating that the main agent mixed with adjuvant would prolong the residue time of imazapyr in the soil to a certain extent. The residue values of Ib and other samples decreased significantly on the 1st and 6th days, respectively, after spraying ($p < 0.05$). Table 2 shows that plant density is significantly positively correlated with TC and TOC ($p < 0.01$) and significantly negatively correlated with EC ($p < 0.01$), while the residue is only significantly positively correlated with EC ($p < 0.05$).

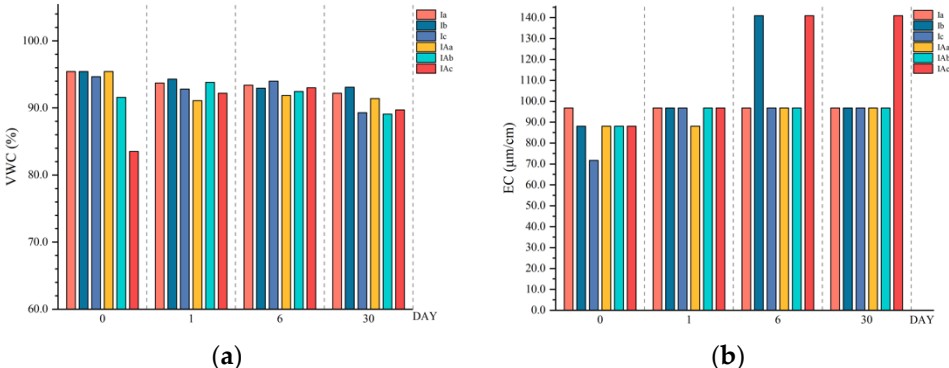

(**a**)　　　　　　　　　　　　　　　　　(**b**)

**Figure 3.** *Cont.*

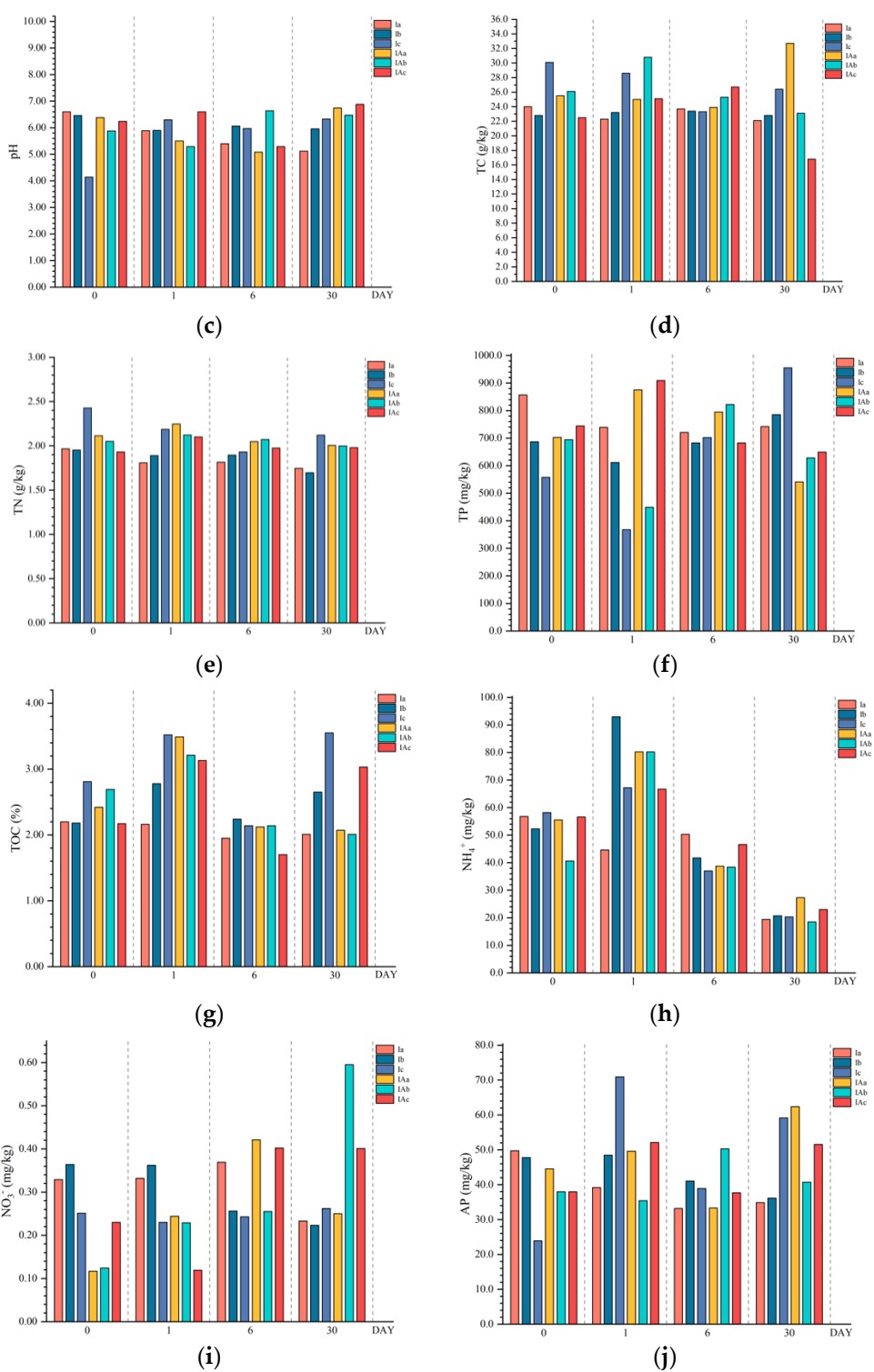

**Figure 3.** Changes in soil physicochemical properties at different plant growth stages: (**a**) soil VWC; (**b**) soil EC; (**c**) soil pH; (**d**) soil TC; (**e**) soil TN; (**f**) soil TP; (**g**) soil TOC; (**h**) soil $NH_4^+$-N; (**i**) soil $NO_3^-$-N; (**j**) soil AP.

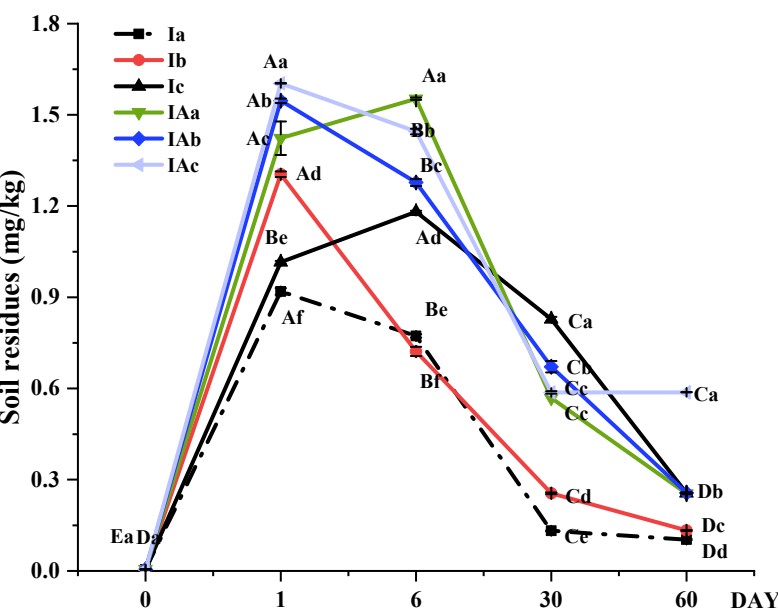

**Figure 4.** Pesticide residues in soil. The different lowercase letters indicate significant differences between the treatment groups ($p < 0.05$), and the different uppercase letters indicate significant differences between time stages ($p < 0.05$).

**Table 2.** Correlation coefficients between residues and soil physicochemical properties and plant density.

|  | VWC | EC | PH | TC | TN | TP | TOC | AP | NH$_4^+$-N | NO$_3^-$-N |
|---|---|---|---|---|---|---|---|---|---|---|
| Plant density | −0.222 | −0.609 ** | −0.158 | 0.634 ** | 0.209 | 0.274 | 0.716 ** | 0.118 | −0.182 | 0.471 |
| residues | −0.071 | 0.568 * | 0.27 | −0.363 | −0.251 | −0.12 | −0.299 | 0.171 | −0.343 | −0.028 |

\* significant at 0.05 level, ** significant at 0.01 level.

Generally, herbicide residues in the environment are a prerequisite for measuring widespread herbicide use. The irrational use of herbicides may cause adverse effects such as farmland waterbody, soil, and air pollution [20,28]. It has been reported that the half-life of imazapyr in soil is relatively long, varying from 20 to 80 days depending on its type. It has the longest half-life in fluvo-aquic soil [29]. According to Figure 3 and Table 2, the impact of plants on soil in the later stage was greater than that of imazapyr in the earlier stage. It may be due to the fact that the amount of imazapyr entering the soil was small and that it was degrading continuously, and that the reduction of plants directly and significantly changes TC and TOC, leading to the change of bacterial abundance and communities; and bacteria change the available soil nutrients through processes such as decomposition, mineralization, nitrification, and denitrification so as to form a cycle mechanism [30]. Furthermore, several studies have shown that imazapyr has little toxicity and is almost nontoxic to aquatic organisms [31–34]. Patten's [33] regular environmental risk assessments and toxicity tests of imazapyr administered annually to *S. alterniflora* showed no risk and toxic effects on local aquatic organisms, birds, and insects. Therefore, as per Patten [33], the herbicide was low-risk within the dose range of this study, even without considering the residue. However, the degradation time of imazapyr was prolonged by the high adjuvant dose. Hence, following the principle of environmental risk minimization, high adjuvant doses should be avoided when using imazapyr. During the regular monitoring of the samples, we observed that the mangrove plants in the quadrat did not exhibit abnormal phenomena such as yellowing of leaves and blocked development, indicating that imazapyr (within the dose range of this study) had no side effects on mangrove plants.

### 3.3. Effects of Imazapyr and Its Additives on Soil Bacterial Abundance and Diversity

The species accumulation curve reflects the effects of the sampling number on species diversity. The gentle rising trend at the end of the curve shows that the sample quantity is enough and that the soil microbial diversity cannot be increased by adding more samples. The results show (Figure 5) that the curve finally flattens out, indicating that the samples in this study are enough to reflect the diversity of soil microbial species.

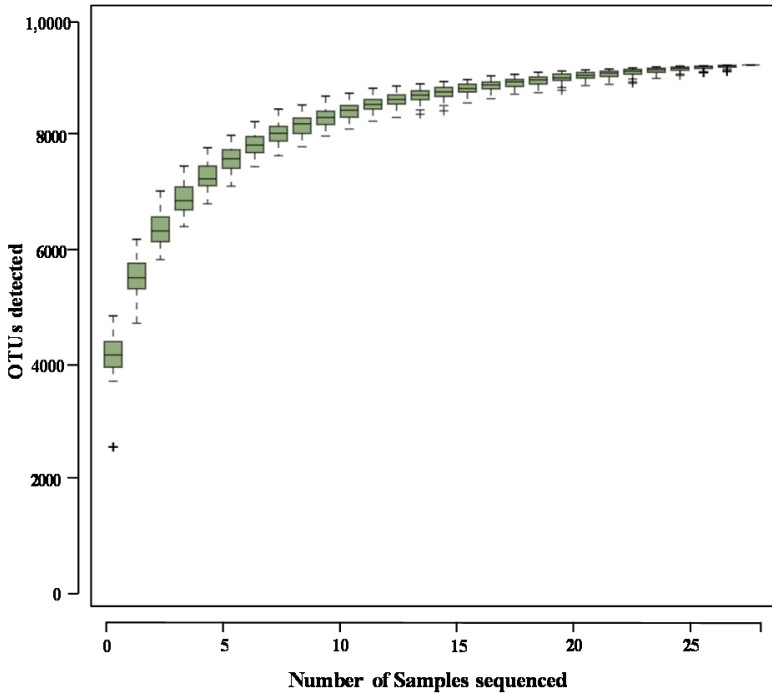

**Figure 5.** Species accumulation curve.

The V3–V4 region of the 16S rDNA gene was sequenced from 18 samples under different treatments, resulting in a total of 776,488 high-quality sequence readings. Among the different treatment groups, the highest sequence quantity appeared in the Ic sample (Table 3; $p < 0.05$). There was no significant difference among the other treatment groups. Regarding different periods, all soil samples showed initially decreasing and then rising trends. Still, there were significant differences only in Ib, Ic, and IAb ($p < 0.05$), which may be related to the degradation of imazapyr. The pesticides not absorbed by plants entered the soil, which provided rich nutrients for microbial growth and metabolism. Then, with the degradation of imazapyr, the number of bacteria decreased. Thereafter, the concentration of the main agent in the soil began to stabilize, the bacteria gradually formed a more stable community structure, and their number finally recovered gradually. The sequences were divided into different OTUs according to the 97% similarity threshold, and each OTU was regarded as a microbial species. The OTU classifications showed that the number of OTUs in IAb and IAc was significantly higher than that in Ic (Table 3; $p < 0.05$). After 30 days of treatment with different doses of adjuvant, the OTUs decreased significantly with increased doses. In addition, the effect of the herbicide can be more attributed to the death of *S. alterniflora* in the later stages induced by imazapyr's inhibition of DNA synthesis, and the production of branched-chain aliphatic amino acids required for plant growth. Finally, the soil vegetation coverage and the plant rhizosphere secretion input decreased.

**Table 3.** Analysis of the microbial diversity indices of the soil samples.

| | Tag Number | | OTU Number | | Chao1 | | ACE | | Shannon | | Simpson | |
|---|---|---|---|---|---|---|---|---|---|---|---|---|
| Ia1 | 43,745 ± 3597.43bc | | 4395.67 ± 141.21fg | | 6636.99 ± 229.62e | | 8051.70 ± 22,770cd | | 6.80 ± 0.026g | | 0.0035 ± 0.00009d | |
| Ia6 | 38,803.33 ± 1069.46c | B | 4784.67 ± 47.26de | AB | 7534.57 ± 170.51bc | A | 9228.64 ± 188.63ab | AB | 7.03 ± 0.041d | A | 0.0027 ± 0.00016g | BC |
| Ia30 | 42,417 ± 2856.91c | | 4749 ± 85.06de | | 7267.80 ± 133.38c | | 8440.74 ± 958.35bc | | 6.95 ± 0.015ef | | 0.0031 ± 0.00012ef | |
| Ib1 | 47,514 ± 3408.79b | | 4480 ± 39.05f | | 6560.89 ± 164.63e | | 7595.95 ± 917.97d | | 6.73 ± 0.026h | | 0.0042 ± 0.00008b | |
| Ib6 | 39,802.67 ± 2017.67c | B | 5172.67 ± 41.96b | AB | 7791.87 ± 143.42a | AB | 8988.58 ± 974.66bc | B | 7.15 ± 0.012b | AB | 0.0025 ± 0.00002h | B |
| Ib30 | 39,864.67 ± 1915.39c | | 4252.33 ± 31.39g | | 6396.81 ± 204.36e | | 7664.25 ± 272.59cd | | 6.84 ± 0.023g | | 0.0036 ± 0.00012d | |
| Ic1 | 54,614 ± 2091.4a | | 2971.67 ± 34.06h | | 3902.45 ± 127.42f | | 4048.71 ± 153.72f | | 6.13 ± 0.014i | | 0.0075 ± 0.00012a | |
| Ic6 | 40,877 ± 778.77c | A | 4964 ± 32.19cd | B | 7597.27 ± 148.29b | B | 9102.58 ± 85.52b | C | 7.05 ± 0.015d | B | 0.0028 ± 0.00006fg | A |
| Ic30 | 47,561.33 ± 1875.13b | | 4822.67 ± 109.18d | | 7168.43 ± 122.28cd | | 7203.39 ± 155.02d | | 6.96 ± 0.020ef | | 0.0033 ± 0.00013e | |
| IAa1 | 43,478 ± 5648.47bc | | 4646.67 ± 102.72e | | 7150.67 ± 141.16cd | | 8569.66 ± 93.28bc | | 6.98 ± 0.023e | | 0.0027 ± 0.00016g | |
| IAa6 | 42,171 ± 1547.36c | B | 4806 ± 167.52de | AB | 7410.37 ± 274.34bc | A | 8989.56 ± 423.11bc | AB | 6.98 ± 0.025e | A | 0.0028 ± 0.00006fg | BC |
| IAa30 | 41,204.67 ± 1845.65c | | 4537.67 ± 37.17ef | | 6943.70 ± 51.28d | | 8416.68 ± 204.54bc | | 6.93 ± 0.037f | | 0.0031 ± 0.00027ef | |
| IAb1 | 51,154.67 ± 3123.97ab | | 4601.67 ± 49.52ef | | 6525.03 ± 229.82e | | 6731.94 ± 203.99e | | 6.80 ± 0.019g | | 0.0039 ± 0.00016c | |
| IAb6 | 38,562.67 ± 1270.07c | B | 5542.67 ± 15.17a | A | 8030.69 ± 85.36a | A | 8370.64 ± 126.30c | B | 7.30 ± 0.027a | A | 0.0022 ± 0.00009i | BC |
| IAb30 | 40,284 ± 2776.52c | | 4659.67 ± 152.68e | | 7292.71 ± 116.20c | | 8820.81 ± 279.42bc | | 6.96 ± 0.041ef | | 0.0029 ± 0.00014f | |
| IAc1 | 40,916.33 ± 120.38c | | 5042.33 ± 120.38c | | 7656.86 ± 145.03b | | 9470.94 ± 140.69ab | | 7.11 ± 0.026c | | 0.0024 ± 0.00011hi | |
| IAc6 | 39,968.67 ± 1809.1c | B | 5241.33 ± 143.20b | A | 8003.24 ± 239.65a | A | 9835.43 ± 317.38a | A | 7.19 ± 0.050b | A | 0.0022 ± 0.00015i | C |
| IAc30 | 43,549.33 ± 3172.85bc | | 4955.67 ± 131.96cd | | 7381.83 ± 271.23bc | | 8824.52 ± 208.37bc | | 7.04 ± 0.007d | | 0.0027 ± 0.00001gh | |

The different lowercase letters in each column of the table mean significant differences among different soil samples, and the different capital letters mean significant differences among the different treatment groups according to the least significant difference (LSD) multiple range test ($p < 0.05$).

The soil bacterial diversity in this study was evaluated using species richness indices (Chao1 and ACE) and diversity indices (Shannon and Simpson). The results (Table 3) showed that the trends of the Shannon, Chao1, and ACE indices for different treatment groups were consistent with the OTUs; the mean value was the lowest in Ic, whereas the highest mean value was in IAc ($p < 0.05$). These results were contrary for the Simpson index. The Shannon, Chao1, and ACE indices all decreased between 1 and 6 days after imazapyr was added with no significant difference ($p > 0.05$), but IAC's ACE index was significantly higher than IAb's. These results indicated that the adjuvant addition promoted microbial diversity, which was inhibited by the initial concentration of imazapyr. In addition, the Shannon index of IAa, the Chao1 and ACE indices of IAa, and the ACE index of IAc in all samples increased significantly on the 6th day after spraying ($p < 0.05$) but decreased in different degrees on the 30th day. Conversely, the trend of the Simpson index was the opposite. This may be because imazapyr increased the diversity of soil nutrients, which resulted in the emergence of some specific microorganisms and enhanced the competitiveness of bacterial communities dominated by imazapyr-like metabolites.

The effects of 11 soil factors on the overall characteristics of soil microbial diversity under different treatments were analyzed using a Monte Carlo test. The results showed the following order of importance ($F$): TOC > TP > AP > $NH_4^+$-N > TN > TC > EC > $NO_3^-$-N > soil residues > VWC > pH (Table 4). Among these, TOC reached a very significant level ($p < 0.01$), TP, AP, $NH_4^+$-N, and TN reached a significant level ($p < 0.05$), whereas soil residues had no significant effect. These indicate that bacterial diversity was not sensitive to the range of residues in the study and TOC, TP, AP, $NH_4^+$-N, and TN were the main factors, which might be due to the decomposition of chemicals and the reduction of *S. alterniflora* coverage, resulting in the changes of soil physicochemical properties [35]. It is well known that C, N, and P are the key factors affecting soil bacterial metabolism, influencing the nutrient source of bacteria [30]. After 30 days, the nutrient source pathway decreased with the decomposition of imazapyr in the soil, and the microbial abundance thus decreased.

**Table 4.** Importance ranking and significance of soil factors in their explanation of the overall bacterial diversity of all soil samples.

| Soil Factors | Importance Ranking | Explanatory Quantity | $P$ | $F$ |
|---|---|---|---|---|
| TOC | 1 | 47.6 | 0.002 | 14.546 |
| TP | 2 | 27.7 | 0.024 | 6.141 |
| AP | 3 | 22.8 | 0.03 | 4.721 |
| $NH_4^+$-N | 4 | 20.9 | 0.042 | 4.227 |
| TN | 5 | 18.3 | 0.048 | 3.583 |
| TC | 6 | 11.5 | 0.19 | 2.078 |
| EC | 7 | 5.2 | 0.352 | 0.881 |
| $NO_3^-$-N | 8 | 3.8 | 0.47 | 0.626 |
| Soil residues | 9 | 3.6 | 0.496 | 0.605 |
| VWC | 10 | 0.6 | 0.0818 | 0.092 |
| pH | 11 | 0.3 | 0.091 | 0.04 |

Vegetation is another key factor affecting soil bacterial communities [35]. The root exudates of plants directly change soil properties, and they can directly or indirectly create suitable living environments for soil microorganisms [35,36]. This increases soil complexity and, in turn, soil microbial diversity. With the action of the imazapyr, plant growth and metabolism were inhibited, and rhizosphere secretion was reduced. Given the open environment of the Zhangjiang Estuary Wetland, most of the dead and fallen plants were transported long distances with the tide movement, hampering their accumulation in the sediment surface for decomposition. The soil C, N, and P input sources were reduced, resulting in significant changes in microbial abundance. Furthermore, some specific microbial communities were reduced, thus simplifying the soil microbial structure [36]. The microbial communities that could adapt to the changing vegetation environment gradually

became the dominant community, finally leading to changes in soil bacterial diversity and structure. This was consistent with the change trend of soil bacterial species richness and diversity in this study ($p < 0.05$).

Patten previously studied the impact of the imazapyr treatment of *S. alterniflora* on the surrounding environment and organisms and found that imazapyr had no environmental risk and was almost nontoxic to organisms [20]. In the study, the soil residues decreased in very significant trends from Days 6 to 30, consistent with Patten's findings, and the residues had no significant effect on bacterial diversity (Table 4) indicating that the imazapyr of the study did not pose a threat to bacteria in the short term. Therefore, the first consideration after controlling *S. alterniflora* should be to fill up the vacated ecological niche to prevent the secondary invasion of *S. alterniflora* and restore soil microbial diversity and the balance of the local ecosystem.

### 3.4. Effects of Imazapyr and Its Additives on the Structure of Soil Bacterial Communities

Figure 6 shows the relative abundance of the top 10 bacterial phyla in total samples, with an average total relative abundance greater than 0.3%: Proteobacteria ($52.91 \pm 1.94\%$), Bacteroidetes ($10.48 \pm 3.00\%$), Acidobacteria ($9.01 \pm 1.62\%$), Chloroflexi ($6.77 \pm 2.10\%$), Actinobacteria ($3.19 \pm 1.09\%$), Firmicutes ($1.61 \pm 0.66\%$), Nitrospirae ($0.95 \pm 0.26\%$), Verrucomicrobia ($0.60 \pm 0.15\%$), Ignavibacterium ($0.57 \pm 0.21\%$), and Cyanobacteria ($0.39 \pm 0.29\%$). The LSD test showed that Proteobacteria, Bacteroidetes, Acidobacteria, Chloflexi, and Actinobacteria were significantly more abundant than were the other phyla ($p < 0.05$), among which Proteobacteria was the most abundant ($p < 0.01$). Therefore, these phyla dominated the soil samples, with the absolute predominance of Proteobacteria. These results were consistent with Cui et al. [37]. Generally, these phyla are widespread in various soil types and are the most common microbial communities in soils with high metabolic diversity [38–41]. They play key roles in maintaining the balance of material and energy cycles in ecosystems through metabolic processes such as C fixation, N fixation, decomposition, nitrification, and denitrification [42]. Moreover, Ignavibacterium, a denitrifying bacteria, have been extensively studied for their ability to denitrify and oxidize sulfides and also affect methane emissions from wetlands [43,44]. They are a common community of coastal marsh wetlands, and their abundance and membership often change after *S. alterniflora* invasions [44].

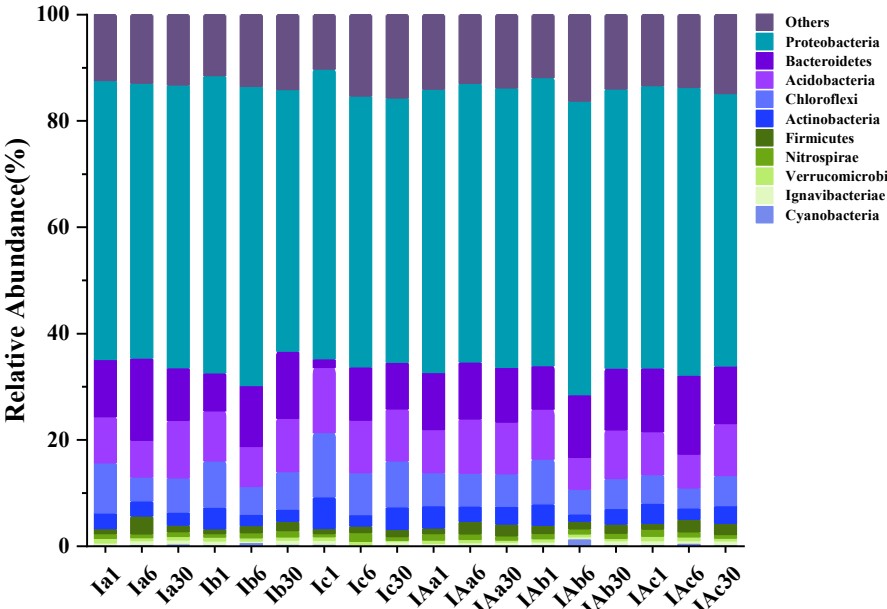

**Figure 6.** Relative abundances of bacterial compositions on the phylum level.

There were no significant differences in the relative abundance of Proteobacteria between Ia, Ib, and Ic ($p > 0.05$). The relative abundances of Acidobacteria, Chloroflexi, and Actinobacteria in Ic soil were significantly higher than those in Ia and Ib ($p < 0.05$), whereas for Bacteroidetes it was the reverse ($p < 0.05$); there was no significant difference between IAa, IAb, and IAc ($p > 0.05$). Meanwhile, Acidobacteria, Chloroflexi, and Actinobacteria in Ic soil were significantly higher than those in IAc but significantly lower than those in IAb and all adjuvant treatments. These results are consistent with the findings of some studies that the use of herbicides does not significantly affect the distribution of soil microbial communities on a large scale in the short term [45,46]. Moreover, Acidobacteria, Chloroflexi, and Actinobacteria are oligotrophic bacterial communities widely distributed in soil [41]. In contrast, Bacterioidetes are eutrophic bacteria sensitive to carbohydrates, lipids, acids, and amine carbon sources [35]. These explain the significant changes in these colonies in these results.

At different time stages, the relative abundance of Proteobacteria only changed significantly in Ib and Ic ($p < 0.05$), and that of Bacteroidetes markedly increased on Day 6 in IAa, Ib, Ic, IAb, and IAc ($p < 0.05$). The relative abundances of Acidobacteria and Chloroflexi decreased significantly on Day 6 in all samples except for IAa. The relative abundance of Actinobacteria decreased significantly from Days 6 to 30 in Ia, and those in all other samples decreased significantly from Days 1 to 6. Some of these dominant soil bacterial communities under the *S. alterniflora* were intolerant to imazapyr. In contrast, the others had universal adaptability to environmental stress and primarily decomposed imazapyr. The Bacterioidetes were the most resilient and have been found to have a certain ability to remediate soil pollution [39]. Therefore, communities of Bacteroidetes can adapt to new soil environments quickly, even after spraying imazapyr. Moreover, imazapyr had low availability to other communities, which reduced the food source competition of Bacteroidetes.

The PCoA results based on the weighted-UniFrac distance to the community structure and the permutational multivariate ANOVA (PERMANOVA) test using the R software showed that the first two principal components accounted for 63.83% of the data matrix variance, and the I and II axes were 48.93% and 14.9%, respectively (Figure 7). These clearly show no significant separation between the different treatment groups and that there was no significant difference in the distance between the groups detected by PERMANOVA ($R^2 = 0.275$, $p = 0.914$). These results indicate that there was no significant difference in the overall structure of soil bacteria treated with different doses of main and auxiliary agents based on the level of community abundance change.

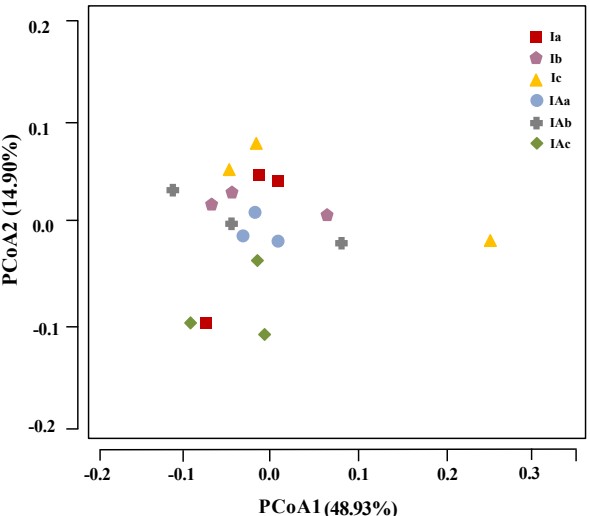

**Figure 7.** PCoA based on the weighted-UniFrac distance to the structures of bacterial communities at the different treatments.

The results of the RDA (Figure 8) show that the explanatory capacities of the main soil bacterial communities (top 10 phyla) under different treatments were 67.7% and 10.1%, respectively, on the I and II axes. That is, the first two ranking axes of the 10 soil physicochemical properties and imazapyr residues cumulatively explain 77.8% of the changes in the main bacterial community. These results show that the first axis mainly reflects the distribution characteristics of the main soil bacterial communities and most information on soil factors in the study area. Among these, TOC, AP, and TP were the most influential factors on the overall characteristics of the main soil bacterial communities in the different treatments. The Monte Carlo test results further proved that TOC, AP, and TP were extremely significant (Table 5, TOC, $p < 0.01$) and significant (AP and TP, $p < 0.05$) influencing factors. Studies have shown that C, N, P were the main factors that affect the soil microbial communities after the invasion of *S. alterniflora*, which further affects the C and N cycle of wetland ecosystem [47,48]. Meanwhile, the soil residues also had no significant impact on the main bacterial communities as a whole. Studies have shown that pesticide residues may pose a potential threat to the soil environment and harm soil microorganisms. Thus, the impact of pesticide residues on soil microorganisms can be used as a basis for the application of herbicides [46]. Meanwhile, the abundances of Acidobacteria, Chloroflexi, and Actinobacteria in Ic were significantly higher than those in IAc ($p < 0.05$) but significantly lower than those in IAb and all adjuvant treatments ($p < 0.05$). At different time stages, the relative abundance of Proteobacteria changed significantly only in Ib and Ic ($p < 0.05$), and that of Bacteroidetes increased significantly on Day 6 in Ia, Ib, Ic, IAb, and IAc ($p < 0.05$). These communities with the ability to degrade chemicals might have played major roles in degrading imazapyr [35], whereas only TN was significantly different among the different treatments. These results further indicated that the main driving factors that caused the changes in the dominant communities at the genus level are the changes in vegetation characteristics and imazapyr itself (including the addition and degradation), with vegetation being the most influential factor.

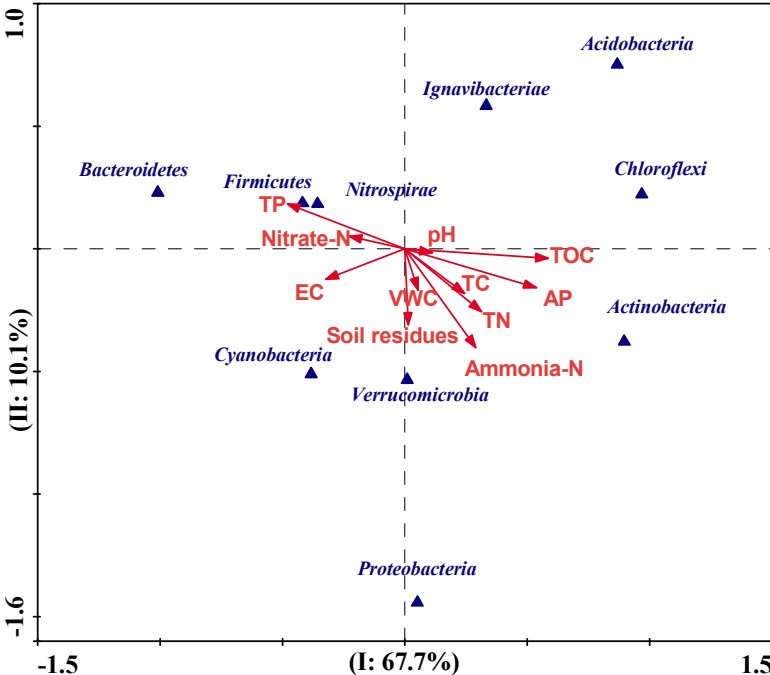

**Figure 8.** Redundancy analysis (RDA) between the 10 dominant phyla and 11 soil factors.

**Table 5.** Importance ranking and significance of soil factors in their explanation of the overall bacterial dominant communities in all soils.

| Soil Factors | Importance Ranking | Explanatory Quantity | $P$ | $F$ |
|---|---|---|---|---|
| TOC | 1 | 29 | 0.004 | 6.525 |
| AP | 2 | 24.9 | 0.024 | 5.299 |
| TP | 3 | 20.7 | 0.026 | 4.166 |
| $NH_4^+$-N | 4 | 12.6 | 0.108 | 2.296 |
| TN | 5 | 10.9 | 0.128 | 1.948 |
| EC | 6 | 9.4 | 0.172 | 1.652 |
| TC | 7 | 6 | 0.314 | 1.028 |
| $NO_3^-$-N | 9 | 4.8 | 0.396 | 0.807 |
| Soil residues | 8 | 3.4 | 0.582 | 0.571 |
| VWC | 10 | 2.7 | 0.66 | 0.446 |
| pH | 11 | 1.4 | 0.872 | 0.232 |

## 4. Conclusions

Within the experimental dose range in this study, imazapyr's efficiency in controlling *S. alterniflora* reached 90% in all samples, and that of the sample with a medium-dose main agent and auxiliary agent was the highest (95.9%). With half-lives less than 30 days in plants and soil, all imazapyr residues in the samples dropped to a low value after the 60th day. Imazapyr residues had no significant effects on mangrove plants, and the change of bacterial diversity and community structure were mainly attributed to the addition and degradation of chemicals and the reduction of *S. alterniflora* coverage. Therefore, considering the principle of environmental and higher effective control efficiency, the combination of 1299 mL/acre AGE 809 + 6070 mL/acre 25% imazapyr is the best formulation for large-scale use in mangrove wetland reserves and coastal wetlands of China. This study is of great significance for evaluating the risk of using herbicides on soil environments and bacteria in the short term and provides a scientific basis for relevant bodies to formulate policies for controlling *S. alterniflora*. Furthermore, it provides a theoretical basis and data and technical support for controlling *S. alterniflora* through herbicide management in the coastal wetlands of China.

**Author Contributions:** Conceptualization, X.M., M.Z. and Z.Z.; methodology, X.M., J.L.; software, X.M.; validation, M.Z., Z.Z.; formal analysis, X.M.; investigation, P.D., L.X., X.S., Y.X. and Y.W.; resources, M.Z., B.W.; data curation, X.M.; writing—original draft preparation, X.M., P.D., L.X., Y.X. and Y.W.; writing—review and editing, X.M.; visualization, X.M.; supervision, M.Z., Z.Z.; project administration, M.Z., Z.Z.; funding acquisition, M.Z., Z.Z. and B.W. All authors have read and agreed to the published version of the manuscript.

**Funding:** This research received no external funding.

**Data Availability Statement:** Yes, please contact the first author at mxoue998@163.com.

**Acknowledgments:** The work described in this paper was supported by Special Project of Forestry and Grassland Science and Technology Innovation, Development and Research of the State Forestry and Grassland Administration (2020LCKCZ005).

**Conflicts of Interest:** The authors declare no conflict of interest.

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
