# Peer review of "Effects of Imazapyr on Spartina alterniflora and Soil Bacterial Communities in a Mangrove Wetland"

_water, doi:10.3390/w13223277_

Round 1

Reviewer 1 Report

The manuscript entitled “ Effects of imazapyr on Spartina alterniflora and soil bacterial communities in a mangrove wetland” is interesting but needs a major revision before acceptance.

Here it goes some ideas and comments you may select to improve your work.

“After 10 years of extensive use of imazapyr on its west coast, the United States has achieved very effective results. It has controlled the invasion of S. alterniflora, and subsequent environmental monitoring studies showed that the use of the herbicide had no risk to the surrounding environment and organisms."

If these works have been performed almost 20 years ago (and many others can be read), what are the differences and the questions you raised in this work experiment? You should avoid saying that this is important to check in China. Yes, it is, but you need to have a scientific question and apparently you didn’t mention it. 

Also, there are numerous articles focus on soil microorganisms and how they are affected by different environmental as well as human impacts. What is your adding value to the science of salt-marsh vegetation and ecology? 

One of the questions you inserted at the end of your introduction was:

“to screen the best formula of imazapyr with high efficiency, economy, and safety”. High efficiency you said something but what in terms of economy and safety? You said no word about this. Safety in terms of soil microorganisms ? But you check that most part of them are so dependent of plants… so, if you arrange a way to take out plants soil micro-organism will also disappear. Also safety is generally uses in terms of human health safety, if it is not the case please rephrase it.

If you want to give a broader idea about the applied importance of this study you should give comparative values of cost efficiency and safety for populations. Spraying is manual behaviour I imagine is not so cheap and is time dependent. 

There are no words about tides and you may know that many plants are limited by the upper high tide water level. You said nothing about this. Please explain. 

All these aspects are important since you sprayed your plants with imazapyr. In what part of the day and how much of this herbicide is washed out by the tide?

Why did you use AGE 809 (Lecithin) ? You mentioned only imazapyr not this one. Explain the reason why you choose this and why you added those quantities. Also, explain why you choose those concentrations of imazapyr. Based on what?

What do you mean “The spraying frequency covers the day before spraying “ Line 134 ? Frequency is the number of days you spray. If you sprayed the day before…before of what? At the very end, how many following days did you spray your plants? It was a daily spray? Thinking on tides did you spray before or after high tide, or didn’t you take care on that?

Soil samples were randomly collected …. along the strip ? Parallel or perpendicular? As far as I understood from your scheme Fig 1, the two sites (1 and I) are parallel, and perpendicular to the estuary. However the 3 strips in each are also parallel but also parallel to the estuary. The effect of tide are completely different under these conditions. Did you take this into consideration?

I understand that you choose bacteria as your main group of soil microorganisms. I also know that only few fungi can appear but there are many actinomycetes that you should have not forgotten. Why you didn’t look those organisms?

In Figure 3 why you didn’t show statistical differences? You should.

In lines 331-334 you said that there was a diminishing complexity of soil microorganisms with time due to the efficient effect of herbicide on vegetation pattern. It’s a pity you didn’t show that evolution. On the contrary, apparently, you showed significant correlation between soil chemical factors and bacterial dominance. This type of relation is not precise because in top of that are the plants or their absence that regulate the most part of all those factors you have used. So, you should take this into consideration and compare these values with others from mangrove or even from other “normal” Spartina species (if you still have). 

Saying this, I don’t know if the addition of a sophisticated evaluation of microorganisms is worthwhile for the major conclusion you take as “main driving factors for the changes in soil bacterial community diversity and structure in the mangrove wetland were the coverage and mortality of S. alterniflora.” But you add that based only in your experiments not in comparable situation in another part of the marsh. At least some references you should add. Do you have the same proportion of groups in different sites? are the soil factors similar to the ones in another parts without this invasive species? Looking at values of Soil P or NH4 (major source of N) are these explaining the great abundance and invasiveness of this species? All these are responsible for certain group of bacteria, for instances Proteobacteria that varies significantly, include heterotrophs, autotrophs, and methanotrophs. The best known heterotrophs species probably play a major role in carbon turnover. Members of Burkholderia are also reported to fix nitrogen and promote plant growth. The others of N cycle - cyano- and actinobacteria - also vary significantly. These means that those important groups have been affected by plant disappearance or mainly by the effect of herbicide?

I understand your convenience in presenting results and discussion in only one section but I would advise to add a better and more holistic conclusion. 

Author Response

Dear Reviewer:

We are very grateful to you and your patience in commenting on and making suggestions for our manuscript (ID: water-1444626). We have made corresponding revisions according to the reviewer’s comments. All revisions to the manuscript had been marked up using the “Track Changes” function. we tried our best to improve the manuscript. Now, the revised manuscript has been resubmitted to your journal, we look forward to your positive response. The follows are the response to your comments.

“After 10 years of extensive use of imazapyr on its west coast, the United States has achieved very effective results. It has controlled the invasion of S. alterniflora, and subsequent environmental monitoring studies showed that the use of the herbicide had no risk to the surrounding environment and organisms."If these works have been performed almost 20 years ago (and many others can be read), what are the differences and the questions you raised in this work experiment? You should avoid saying that this is important to check in China. Yes, it is, but you need to have a scientific question and apparently you didn’t mention it. 

Also, there are numerous articles focus on soil microorganisms and how they are affected by different environmental as well as human impacts. What is your adding value to the science of salt-marsh vegetation and ecology? 

Response:Thank you for your suggestions. So far, there is no reports on the use of imazapyr to control S. alterniflora in China, its ability to prevent and control the invasion of S. alterniflora in China and its possible environmental impacts are unclear. In addition, the USA has been studying the effect of imazapyr removal of S. alterniflora on water, non-target plants, birds, in-sects, fish) for many years, but did not involve the effects on soil physicochemical properties and microorganisms. In order to analyze the control effect of imazapyr and its mixture with adjuvant on S. alterniflora of Chinese wetland, and evaluate the effect of imazapyr on soil, especially on soil microbial communities and dominant factors

Comment 1: “to screen the best formula of imazapyr with high efficiency, economy, and safety”. High efficiency you said something but what in terms of economy and safety? You said no word about this. Safety in terms of soil microorganisms ? But you check that most part of them are so dependent of plants… so, if you arrange a way to take out plants soil micro-organism will also disappear. Also, safety is generally uses in terms of human health safety, if it is not the case please rephrase it.

Response: The chemical method is cheaper than the physical treatment method, thereby this paper measures the economic standard by the amount of reagent added, but after considering your comment, this content has been deleted in the manuscript. “safety” has been modified to "low-risk", during the regular monitoring of the samples, we observed that the mangrove plants in the quadrat did not exhibit abnormal phenomena such as yellowing of leaves and blocked development, indicating that imazapyr (within the dose range of this study) had no side effects on mangrove plants.

Comment 2: If you want to give a broader idea about the applied importance of this study you should give comparative values of cost efficiency and safety for populations. Spraying is manual behaviour I imagine is not so cheap and is time dependent. 

Response: In this study, the experimental area is small, and the UAV is suitable for large area, and it is not suitable for small area experiment, so the artificial spraying is the best choice.

Comment 3: There are no words about tides and you may know that many plants are limited by the upper high tide water level. You said nothing about this. Please explain.  All these aspects are important since you sprayed your plants with imazapyr. In what part of the day and how much of this herbicide is washed out by the tide?

Response: The experimental area is located in the intertidal tidal flat, submerged by the diurnal tide, and the plant community in the area is completely submerged during high tides and 5 ~ 8 hours is the low tide. During the low tide, the sea water can not submerge the leaves of S. alterniflora. The sampling time covers the day before spraying and days 15, 30, and 60 after spraying imazapyr, samples were collected after tide-ebbing. Furthermore, the main uptake time of the imazapyr by the plant is within a few hours after the ebb tide, and most of the imazapyr without adjuvant will migrate with the water after the next high tide, but this is enough to kill most of the S. alterniflora. See section “2.1 Overview of the study area”.

Comment 4: Why did you use AGE 809 (Lecithin)? You mentioned only imazapyr not this one. Explain the reason why you choose this and why you added those quantities. Also, explain why you choose those concentrations of imazapyr. Based on what?

Response: The medicine we used was imported from the United States, and the adjuvant is incidental. When imidazole nicotinic acid is used to treat Spartina alterniflora in the United States, adjuvants are default and must be added. This is because S. alterniflora often grows in the intertidal zone, and imidazole nicotinic acid often dissipates with the tide before it has played its best effect. Therefore, adjuvants are necessary to increase the adhesion of the main agent on plants. One reason why the treatment without adjuvant was set up in this study is to evaluate the effect of pure main agent and analyze the difference of efficacy and impact on soil between adjuvant and non-adjuvant. In addition, American experts suggest that the dose is 3035 ~ 6070 ml/acre, so the dose gradient of this study is designed on this basis.

 Comment 5: What do you mean “The spraying frequency covers the day before spraying “ Line 134 ? Frequency is the number of days you spray. If you sprayed the day before…before of what? At the very end, how many following days did you spray your plants? It was a daily spray? Thinking on tides did you spray before or after high tide, or didn’t you take care on that?

Response: I'm sorry, it's my negligence, it’s not “frequency”, but “time”. It has been revised in the manuscript: “The sampling time covers the day before spraying and days 15, 30, and 60 after spraying imazapyr, samples were collected after tide-ebbing”.

Comment 6: Soil samples were randomly collected …. along the strip ? Parallel or perpendicular? As far as I understood from your scheme Fig 1, the two sites (1 and I) are parallel, and perpendicular to the estuary. However the 3 strips in each are also parallel but also parallel to the estuary. The effect of tide are completely different under these conditions. Did you take this into consideration?

Response: Your comment is very correct, but the samples we drew in the intertidal were close, and they were actually subjected to tides with the same conditions.

Comment 7: I understand that you choose bacteria as your main group of soil microorganisms. I also know that only few fungi can appear but there are many actinomycetes that you should have not forgotten. Why you didn’t look those organisms?

Response: As the most abundant and diverse types among microbial communities, the bacterial community can represent the changes of microorganisms to a certain extent. Furthermore, bacteria and actinomycetes are two independent detection systems in the detection platform. Considering the financial problem, only bacteria module is selected.

Comment 8: In Figure 3 why you didn’t show statistical differences? You should.

Response: In the process of collecting samples, we strictly abide by the principle of equal quantity, random and multi-point mixing, but we couldn’t bring back multi-point repeated soil samples due to weather, environment and human factors. Therefore, we finally mix the mixed quadrats collected from the parallel quadrats under each processing again, which reduces the error to a certain extent. And finally, it is carried out on a professional detection platform. There is no parallel value, so it is impossible to carry out statistical difference analysis, but the detection results have high accuracy.

Comment 9: In lines 331-334 you said that there was a diminishing complexity of soil microorganisms with time due to the efficient effect of herbicide on vegetation pattern. It’s a pity you didn’t show that evolution. On the contrary, apparently, you showed significant correlation between soil chemical factors and bacterial dominance. This type of relation is not precise because in top of that are the plants or their absence that regulate the most part of all those factors you have used. So, you should take this into consideration and compare these values with others from mangrove or even from other “normal” Spartina species (if you still have). åœ¨

Response: I'm sorry, it's my negligence. The correct conclusion would be “a diminishing complexity of soil microorganisms with time due to the addition and degradation of chemicals and the reduction of S. alterniflora coverage”. Plants were another major factor. This study mainly discusses the changes before and after spraying imazapyr in S. alterniflora community. and the impact on mangrove microorganisms is the content to be explored later. According to your comments, Table 2 Correlation coefficients between residues and soil physicochemical properties and plant density and its related results and discussions have been added to the manuscript.

Comment 10: Saying this, I don’t know if the addition of a sophisticated evaluation of microorganisms is worthwhile for the major conclusion you take as “main driving factors for the changes in soil bacterial community diversity and structure in the mangrove wetland were the coverage and mortality of S. alterniflora.” But you add that based only in your experiments not in comparable situation in another part of the marsh. At least some references you should add. Do you have the same proportion of groups in different sites? are the soil factors similar to the ones in another parts without this invasive species? Looking at values of Soil P or NH4 (major source of N) are these explaining the great abundance and invasiveness of this species? All these are responsible for certain group of bacteria, for instances Proteobacteria that varies significantly, include heterotrophs, autotrophs, and methanotrophs. The best-known heterotrophs species probably play a major role in carbon turnover. Members of Burkholderia are also reported to fix nitrogen and promote plant growth. The others of N cycle - cyano- and actinobacteria - also vary significantly. These means that those important groups have been affected by plant disappearance or mainly by the effect of herbicide?

Response: It is really true as your comments. We have revised the manuscript according to your comments and added several documents and related contents. In addition, the conclusion is: the change of bacterial diversity and community structure were mainly attribute to the addition and degradation of chemicals and the reduction of S. alterniflora coverage. The content of S. alterniflora related bacterial community was added to the manuscript. However, this study mainly analyzes the main community of soil bacteria, which can reflect the comprehensive changes of soil bacteria. In fact, we originally want to discuss functional communities such as heterotrophs, autotrophs, and methanotrophs, but this requires more in-depth analysis, which cannot be completed under the limitation of the length of the article, so there is less discussion in this regard.

I understand your convenience in presenting results and discussion in only one section but I would advise to add a better and more holistic conclusion.

Special thinks to you for your good comments! we tried our best to improve the manuscript according to your suggestions.

Reviewer 2 Report

Dear Authors,

I have emailed the review of your manuscript to the Editor.

.

Best regards

Reviewer

Author Response

Dear Reviewer:

We are very grateful to you and your patience in commenting on and making suggestions for our manuscript (ID: water-1444626). We have made corresponding revisions according to the reviewer’s comments. All revisions to the manuscript had been marked up using the “Track Changes” function. we tried our best to improve the manuscript. Now, the revised manuscript has been resubmitted to your journal, we look forward to your positive response. The follows are the response to your comments.

Comment 1: Line 73: complete : …plants, soils, animals, fish, birds and soil microorganisms[23]….

Response: It has been revised in accordance with the comment.

Comment 2: Line 113: „…The soil is mainly semifluid, with oxygen deficiency, a reductive state, and strong acidity!!!. It is rich in plant residues, organic matter, and calcium.” This is not possibility, soil rich in organic matter and calcium will never have high acidity.

Response: "Calcium" has been deleted according to your comment, but the increase of organic matter will increase the content of humus, resulting the reduction of pH.

Comment 3: Line 244: You should complete legend of the figure No. 3. Describe: a, …j.

One should also place homogeneous groups distinguished by, for example, Tukey's test in fig. 3 .

Response: the complete legend has been added in accordance with the comment. In the process of collecting samples, we strictly abide by the principle of equal quantity, random and multi-point mixing, but we couldn’t bring back multi-point repeated soil samples due to weather, environment and human factors. Therefore, we finally mix the mixed quadrats collected from the parallel quadrats under each processing again, which reduces the error to a certain extent. And finally, it is carried out on a professional detection platform. There is no parallel value, so it is impossible to carry out statistical difference analysis, but the detection results have high accuracy.

Comment 4: Line 388: One need also place homogeneous groups in fig.4.

Response: Think you for your comment, the statistical method we used is LSD test, and the results are arranged according to the letter marking method. We think this method is feasible, If you still think it is not feasible, we will modify it.

Special thinks to you for your good comments! we tried our best to improve the manuscript according to your suggestions.

Round 2

Reviewer 1 Report

You still have some missing details that you need to add for a better comprehension and scientific criterion. Please revise your english.

Within your scientific questions: ….on soil microbial communities and dominant factors, 

What do you mean? dominant factors ?

You still didn’t answer to this:

1. All these aspects are important since you sprayed your plants with imazapyr. In what part of the day and how much of this herbicide is washed out by the tide? So, it is important you show in what part of the day you sprayed the plants in order to know how many hours the leaves are in contact with the imazapyr.

2. Why did you use AGE 809 (Lecithin) ? You mentioned only imazapyr not this one. Explain the reason why you choose this and why you added those quantities. Also, explain why you choose those concentrations of imazapyr. Based on what? Why some experiment sites were with AGE and others not?

Table 2 (new one) what is the meaning of the two rows? Please justify . Also, when you add:

“According to Figure 3 and Table 2, the impact of plants on soil in the later stage was greater than that of imazapyr in the earlier stage. It may due to that the amount of imazapyr entering the soil was small and was degrading continuously, and the reduction of plants directly and significantly changes TC and TOC, which directly promote the change of other available nutrients content[32].”

What your data says is that imazapyr affect only C cycle and C soil availability. Apparently where did you see “change of other available nutrient contents?
